# Characterization of Sex-Based Differences in Gut Microbiota That Correlate with Suppression of Lupus in Female BWF1 Mice

**DOI:** 10.3390/microorganisms13051023

**Published:** 2025-04-29

**Authors:** James W. Harder, Jing Ma, James Collins, Pascale Alard, Venkatakrishna R. Jala, Haribabu Bodduluri, Michele M. Kosiewicz

**Affiliations:** Department of Microbiology and Immunology, University of Louisville, Louisville, KY 40202, USA; harderjw89@gmail.com (J.W.H.); jing.ma@louisville.edu (J.M.); james.collins.1@louisville.edu (J.C.); pascale.alard@louisville.edu (P.A.); venkatakrishna.jala@louisville.edu (V.R.J.); haribabu.bodduluri@louisville.edu (H.B.)

**Keywords:** autoimmune disease, systemic lupus erythematosus, gut microbiota, sex, fecal transplant, therapy

## Abstract

Systemic lupus erythematosus (SLE) is more prevalent in female mice and humans and is associated with microbiota dysbiosis. We analyzed the fecal microbiota composition in female and male NZBxNZWF1 (BWF1) mice, a model of SLE, using 16S RNA gene sequencing. Composition of gut microbiota differed between adult disease-prone female (pre-disease) and disease-resistant male mice. Transfer of male cecal contents by gavage into female mice suppressed kidney disease (decreased proteinuria) and improved survival. After our mouse colony was moved to a new barrier facility with similar housing, male cecal transplants failed to suppress disease in female recipients. After two years, the protective phenotype reemerged: male cecal transplants once again suppressed disease in female mice. We compared the gut microbiota composition in female and male BWF1 mice for the three different periods, during which the male microbiota either protected or failed to protect female recipients. In female vs. male mice and in female mice receiving male cecal transplants, we found *Bacteroides* was high, *Clostridium* was low (high *Bacteroides*/*Clostridium* ratio), and *Alistipes* was present during periods when male cecal transplants suppressed disease. These data suggest that specific bacterial populations may have opposing effects on disease suppression in a model of microbiota transplantation.

## 1. Introduction

Dysbiosis, or an alteration in the gut microbiota, is linked to factors such as antibiotic use, diet, infections, and underlying health conditions. It is also frequently associated with autoimmune diseases, including type 1 diabetes, multiple sclerosis, rheumatoid arthritis, and systemic lupus erythematosus (SLE), in both humans and mouse models [1,2,3,4,5]. Sex-based differences in the microbiota have been identified in both humans and mice, with variations in the composition and diversity of microbial communities between the sexes [6,7,8,9]. This is of interest because many autoimmune diseases, including SLE, exhibit a sex bias and are much more prevalent in females than males [10,11]. In SLE, women comprise over 90% of patients. This sex bias is also present in the NZBxNZWF1 (BWF1) mouse model of lupus which spontaneously develops lupus [12], with 100% of female BWF1 mice exhibiting lupus (lupus nephritis) by 34 weeks of age and mortality by 50 weeks of age, whereas male BWF1 mice either do not develop disease or develop much less severe disease and at a much later age [13,14,15].

SLE is a systemic disease that is characterized by dysregulation of the immune response, resulting in production of anti-nuclear antibodies and formation of immune complexes. Deposition of the immune complexes into critical organs such as the kidneys produces chronic inflammation and severe organ damage (e.g., lupus nephritis), and potentially death. Alterations in the gut microbiota have been associated with both SLE patients and mouse models [16,17,18,19,20,21,22,23]. While the specific taxa that are altered in SLE patients vary from study to study, our understanding of how the gut microbiota can affect disease progression has expanded. Compared to healthy controls, higher abundance of *Ruminococcus gnavus* in SLE patients and elevated levels of antibodies against *R. gnavus* have been shown to correlate with overall antibody levels and kidney damage [16]. The presence of *Enterococcus gallinarum* in the liver of SLE patients has been correlated with higher levels of auto-antibodies, while the presence of *Ruminococcus* is associated with decreased Treg levels in SLE patients [24,25]. Similarly, studies using different mouse models of lupus have found different gut bacteria having an impact on disease [19,22,24,26,27], demonstrating a role for gut microbiota in the progression of lupus.

Sex and sex hormone levels also significantly affect the microbiota, suggesting that sex-based differences in microbiota function could potentially contribute to the sex bias of lupus [9,28,29,30]. Two studies have found evidence to support this possibility. In the non-obese diabetes (NOD) mouse model of type 1 diabetes, female mice are more susceptible and male mice are more resistant to disease. However, the studies found that male mice possessed a distinct microbiota that conferred this protection in an androgen-dependent manner [31,32]. In addition, a study using the SWRxNZB F1 (SNF1) mouse model of lupus found differences in microbiota composition between intact male, androgen-depleted male, and female mice [33]. These studies suggest that sex-based differences in microbiota could influence lupus progression, and this is a topic that requires further study. In the present study, we analyzed gut microbiota in the spontaneous BWF1 mouse model of SLE that exhibits a distinct sex bias similar to that found in humans. We found differences in microbial composition in the feces of adult lupus-prone female (pre-disease) and lupus-resistant male BWF1 mice and determined whether transfer of male microbiota to female recipients could protect recipients from kidney disease. Moreover, we took advantage of the impact that a change in animal facility had on shifts in microbiota populations and corresponding changes in the effects of microbiota transfer on kidney disease to identify microbial populations that may confer or facilitate protection, as well as those that may potentially interfere with protection.

## 2. Materials and Methods

### 2.1. Animals and Animal Facility Descriptions

Six-week-old female NZB/BINJ (NZB; #000684) and male NZW/LacJ (NZW; #001058) mice were purchased from The Jackson Laboratory (Bar Harbor, ME, USA) and crossed in our animal facilities (University of Louisville Health Science Campus, Louisville, KY, USA) to produce New Zealand black (NZB) x New Zealand white (NZW) F1 mice (BWF1 mice). Breeders were replaced every 6 months. The BWF1 mice used in this study were housed in two different specific pathogen-free barrier facilities located in two different buildings during three different periods of time on the University of Louisville Health Science Campus (see Appendix A for the timeline described below). Mice were originally housed between the years 2014 and 2016 in the animal facility in the A Tower Building; this is referred to as the A Tower period in this study. Our entire mouse colony was moved to a different animal facility in the Clinical and Translational Research Building (CTRB) in early January 2017 where it currently resides. The initial period (from January 2017 to 2018) after the move to the CTRB is referred to as the CTRB Early period. The later, and more recent, period after the move to the new animal facility is referred to as the CTRB Recent period (from 2019 to present). In this study, we analyzed microbiota composition in the feces from 16-week-old female and male BWF1 mice from the three different animal facilities/time periods: A Tower, CTRB Early, and CTRB Recent. We also tested the ability of male microbiota (i.e., cecal transplants) to protect female BWF1 recipients from the development of kidney disease in the different animal facilities/time periods, and also analyzed the microbiota composition in the cecal transplant recipients.

Mice were maintained in specific pathogen–free barriers (under positive pressure relative to corridors within the facility) in ventilated caging under a 12:12 light cycle with constant temperature (21–23 °C) and humidity (average humidity: 40–45%; humidity range: 35–60%) in both the A Tower and the CTRB animal facilities. All mice in the study were fed the standard Laboratory Autoclavable Rodent Diet 5010 (LabDiet, St. Louis, MO, USA) ad libitum. A summary of the diet composition has been described previously [34], and a more detailed analysis of nutrient content can be found on the manufacturer’s website (LabDiet, St. Louis, MO, USA). Mice were housed using corncob bedding (Animal Specialties and Provisions, Quakertown, PA, USA). The water source during the A Tower (2014–2016) and during the CTRB Early (2017–2018) periods was identical and consisted of 0.2 micron-filtered tap water (from the same tap water source) autoclaved in individual water bottles. The water source during the CTRB period (2019–present) was changed to water that was purified by reverse osmosis (RO) and then autoclaved in individual bottles. All animal procedures were approved by the University of Louisville Institutional Animal Care and Use Committee [for years spanning 2014–present: protocol numbers (dates of approval): 11045 (7 February 2014); 17107 (31 October 2017); 20802 (9 September 2020); 23300 (1 August 2023)].

### 2.2. Animal Handling and Fecal Sample Collection

The mice were housed in an animal barrier as described above. To minimize stress, fecal samples were collected by gently handling each mouse. Mice were briefly placed in a sterile container, and fresh fecal pellets were collected immediately upon spontaneous defecation to avoid contamination. Fecal samples were promptly placed in sterile 1.5 mL Eppendorf tubes and immediately frozen and stored at −80 °C. Samples were collected at a consistent time of day (8:00 a.m.–11:00 a.m.) to reduce variability in microbial composition due to circadian rhythms.

### 2.3. Cecal (Microbiota) Transplant Preparation and Processing

Preparation of 0.03% L-Cysteine HCl Buffer: A stock solution was prepared by dissolving 30 mg of L-cysteine HCl (C6852; Sigma, St. Louis, MO, USA) in 10 mL of 1X PBS. To obtain a 0.03% L-cysteine buffer, 3 mL of the stock solution was diluted in 27 mL of 1X PBS (1:10). The buffer was then filter-sterilized, and 2.5 mL aliquots were dispensed into 15 mL conical tubes for use. L-cysteine was used as a reducing agent to aid in preserving anaerobes [35,36,37,38].

The cecal contents for the microbiome transplant were prepared under aseptic conditions. One 16-week-old male or female BWF1 mouse (average weight: 40 g; range: 38–42 g) used as cecal content donors was euthanized in a biosafety cabinet. An inverted “Y” incision was made along the ventral surface, and the skin was pinned back to expose the peritoneal cavity. The cavity was opened, and the membrane edges were pinned back to reveal the cecum. Once exposed, the cecum was excised using sterile forceps and scissor, then placed on a sterile gauze pad. The cecum was then incised with a scalpel, and its contents were carefully scraped into a conical tube containing 2.5 mL of 0.03% L-cysteine buffer. The suspension was vortexed thoroughly before the addition of 10 mL of L-cysteine, followed by further vortexing to ensure homogeneity. Throughout the procedure, tools were repeatedly sterilized using 70% ethanol and bead sterilizer. The donor cecal content was freshly made at every feeding. Cecal contents were processed and administered to all recipients within 10 min of when the cecum was harvested from the donor mouse.

### 2.4. Cecal Microbiota Transfer

Weanling recipient mice (weight range: 18–19 g) were fed diluted cecal contents from adult 16-week-old female or male BWF1 mice by gavage, according to previously published methods [39], with modifications. Briefly, recipient mice were fed 250 μL of cecal suspension (prepared as described above) via oral gavage using a sterile 1 mL tuberculin syringe (BD, Franklin Lakes, NJ, USA) with a sterile blunt-ended 22-gauge animal feeding needle (Cadence Science, Cranston, RI, USA) as a single dose. Cecal contents were fed successively in the following sequence: Day 1 of feeding, first feeding conducted shortly after weaning (at 22–26 days of age); Day 2, second feeding conducted 24 h after first feeding; Day 8, third feeding conducted 1 week later; Day 36, fourth feeding conducted 4 weeks after last feeding; and then mice were fed every 4 weeks thereafter through the conclusion of the experiment (i.e., severe disease/death). Recipients were randomly assigned to control (female-to-female transfers for all experiments; and male-to-male transfers for some experiments) or treatment groups (male-to-female transfers). Cecal content recipients were monitored for kidney disease (see below for kidney disease scoring), and feces were collected 4 weeks after cecal content feeding and microbiota composition was analyzed (see Section 2.6 for a description).

### 2.5. Measurement of Proteinuria (Kidney Disease Score Assessment)

Kidney disease (glomerulonephritis) was monitored biweekly, beginning at 18 weeks of age until death by measuring proteinuria in the urine. Urine from mice was collected using manual bladder expression. Briefly, mice were gently restrained, and light pressure was applied to the lower abdomen to stimulate urination. Fresh urine was collected directly into sterile tubes to ensure sample integrity and avoid contamination. Proteinuria was assessed semi-quantitatively via a dipstick test using Abustix Urinalysis Reagent Test Strips (Siemens Healthineers, Malvern, PA, USA). The test strips were dipped into fresh urine samples, and results were recorded based on color change following the manufacturer’s instructions. Urine was scored for protein on a scale of 0–4: 0 (none or trace), 1 (30 mg/dL), 2 (100 mg/dL), 3 (300 mg/dL), and 4 (>2000 mg/dL). Kidney disease was considered to be established when proteinuria reached a score of 3 for two consecutive readings after which urine was tested weekly. The incidence of kidney disease was calculated as the percentage of mice with a proteinuria score of ≥3 for two consecutive biweekly readings. A score of 5 indicated death. Mice were euthanized once they reached a body condition score of 2 (of 5) with severe hunching. Proteinuria was evaluated blind by investigators and no animals were excluded from the experiments. The cecal microbiota transfer disease experiments were performed at least 3 times for each time-period (A Tower, CTRB Early and CTRB Recent periods) with similar results (*n* = 5–6; *n* for each experiment is provided in the corresponding figure legend).

### 2.6. Microbiota Composition Analysis and Statistical Analysis

Experimental design: All fecal samples from female and male mice used for microbiota composition analysis were collected from 16-week-old BWF1 mice, including mice housed in the A Tower (2014–2016, *n* = 6–7), CTRB Early Period (2017–2018, *n* = 7–8), and CTRB Recent Period (2019–present, *n* = 7). All fecal samples analyzed for microbiota composition in the cecal transfer experiments were collected at 4 weeks after cecal content transfer from 15/16-week-old (*n* = 6) and/or 19-week-old (*n* = 6) BWF1 recipient mice.

Bacterial DNA from all fecal samples was extracted by QIAamp PowerFecal DNA Kits (Qiagen, Germantown, MD, USA) according to the manufacturer’s instruction. The extracted DNA was sent to the Genome Technology Access Center at Washington University (St. Louis, MO, USA), where nine hypervariable regions of the 16S rRNA gene were sequenced. The resulting multi-amplicon data were then used to generate species-level taxonomic profiles of the microbiota [39]. Briefly, DNA samples were processed using the Fluidigm Access Array System following the manufacturer’s protocol. Amplicon libraries were generated using the Fluidigm Access Array (model LP 48.48 IFC), and sequencing adapters were attached using PCR amplification on the BioMark HD system (Fluidigm, South San Francisco, CA, USA). The pooled samples were purified using AMPure XP (Beckman, Brea, CA, USA) and sequenced on an Illumina MiSeq or HiSeq 3000 instrument. Reads were de-multiplexed based on unique index tags. In our lab, we utilized QIIME2 to compute Bray–Curtis dissimilarity matrices for species-level microbiota compositions and conducted Principal Coordinate Analysis (PCoA). To compare intra- and inter-group Bray–Curtis beta diversity distances, we employed permutational multivariate analysis of variance (PERMANOVA) for pairwise comparisons [40]. PCoAs were visualized using EMPeror 1.17 [41] and plotted in R 3.5.2 [42] using ggplot2 3.3.5 [43] and ggpubr to calculate the 95% confidence ellipse of the group barycenter [44]. Bray–Curtis dissimilarity indices were calculated to assess the differences in microbial community composition within and between the two groups (Group 1 and Group 2). A two-way ANOVA was performed to determine the effects of Group (Group 1, Group 2) and Comparison Type (Within-Group, Between-Group) on Bray–Curtis dissimilarity indices. The independent variables were Group and Comparison Type, while the dependent variable was the Bray–Curtis dissimilarity index. The model used for the analysis was as follows:Distance ~ Group + Comparison Type + Group × Comparison Type

The main effects of Group and Comparison Type, as well as their interaction, were tested for statistical significance. Post hoc analyses were conducted using Tukey’s HSD test (or *t*-test) to identify specific differences between groups and comparison types. Statistical analyses were performed using GraphPad Prism 9, and a significance level of *p* < 0.05 was considered statistically significant.

### 2.7. Statistical Analyses

Disease scores were analyzed using either one- or two-way ANOVA and Student’s *t* test. Incidence and survival curves were assessed using the Log-rank (Mantel–Cox) test. GraphPad Prism version 9 (GraphPad Software, Boston, MA, USA) was used for statistical analysis and assessment of the normality of data; the data were normally distributed. Error bars represent the standard error of the mean (±SEM). A significance threshold of 0.05 was applied, with significance levels denoted as follows: * *p* < 0.05, ** *p* < 0.01, *** *p* < 0.001.

## 3. Results

### 3.1. Sex-Based Differences in Gut Microbiota Composition and Function in the BWF1 Mouse Model of Lupus

In our animal facilities at the University of Louisville, female NZB/NZWF1 (BWF1) mice generally start developing glomerulonephritis by 25–28 weeks of age and virtually all female mice die of this disease by 50 weeks of age. Male BWF1 mice on the other hand only rarely develop a mild form of disease later in life (Appendix A) [13,14,15]. To begin to evaluate the relationship between lupus, gut microbiota, and sex in BWF1 mice, we analyzed the microbiota composition of feces of 16-week-old (pre-disease) female and male BWF1 mice. We used Bray–Curtis dissimilarities to quantify the dissimilarity in microbiota composition within and between groups. The principal coordinate analysis (PCoA) visualization of this analysis showed that the microbiota profiles formed two distinct groups based on sex (Figure 1A). Permutational multivariate analysis of variance (PERMANOVA) of the Bray–Curtis dissimilarities confirmed the female and male BWF1 gut microbiota profiles were significantly different (Figure 1B).

To investigate whether the sex-based differences in the microbiota could affect lupus progression, we carried out microbiota transfer experiments. For these experiments, female BWF1 mice received cecal contents (via gavage) from female (control) or male BWF1 mice, beginning shortly after weaning and continuing monthly throughout the experiment. The transfer of male, but not female, microbiota significantly delayed kidney disease onset and improved survival in female BWF1 mice (Figure 2A–C). Taken together, these data indicate that female and male gut microbiota differ in composition and the male microbiota may contain bacterial populations that can protect against lupus nephritis.

### 3.2. Change in Microbiota Composition Is Associated with Loss of Protective Male Microbiota Transfer Phenotype

The disease progression, and initial microbiota analysis and transfer experiments in BWF1 mice described above, were performed in our specific pathogen-free barrier animal facility in the A Tower building on the University of Louisville Health Sciences Campus from 2014 to late 2016 (A Tower period; Appendix A). In January 2017, our mouse colony was moved to a new barrier animal facility in the Center for Translational Research Building (CTRB). As described in the Section 2, food, water source, bedding, sterilization, husbandry protocols, etc., were similar between facilities, as were disease kinetics, severity, incidence, and mortality in either sex—that is, female BWF1 mice exhibited accelerated and severe glomerulonephritis and increased mortality compared to male BWF1 mice (Appendix A). We then conducted experiments to confirm that male microbiota transferred into female BWF1 mice delayed disease, and decreased severity and mortality in the CTRB animal facility, as we had found in the A Tower. We conducted three microbiota transfer experiments between January 2017 and Fall 2018 (CTRB Early period; Appendix A), but were unable to replicate the protective phenotype that we had seen in the A Tower. We found no difference in the onset and progression of glomerulonephritis between female BWF1 mice that received male microbiota and those that received female microbiota (Figure 3A,B). Survival was also not different between these two groups of recipients (Figure 3C).

We next evaluated the microbiota compositions in the feces of 16-week-old female and male mice (i.e., the microbiota donor populations) from the A Tower and CTRB Early periods to determine whether they remained the same. A Bray–Curtis dissimilarity analysis comparing microbiota composition in feces collected from female and male BWF1 mice from the A Tower and the CTRB Early periods showed that the microbiota compositions differed dramatically between the two periods and the microbiota composition overlapped between female and male mice during the CTRB Early period (Figure 4).

In an attempt to recover our protective transfer phenotype, several changes were made in Fall 2018 to the animal husbandry of our mice in the CTRB facility, which included a change in animal room, a small change in disinfecting protocols used by our animal care technician, and a change in water source, from filtered tap water (i.e., the A Tower water source) autoclaved in water bottles to water purified by reverse osmosis (RO) autoclaved in water bottles. In microbiota transfer experiments performed during this more recent period in the CTRB (i.e., the CTRB Recent period; covering the period of 2019 to the present; Appendix A), disease onset was once again significantly delayed and incidence was decreased, and survival was increased in female BWF1 recipients of male microbiota compared to recipients of female microbiota (Figure 5A–C). Additionally, Bray–Curtis analysis of beta-diversity showed that once again the female and male microbiota profiles form distinct groups (Figure 6A), and were highly significantly different (Figure 6B; *p* < 0.001). Taken together, these data suggest that variability in the ability of male microbiota transplants to protect female mice from disease is associated with alteration in the microbiota composition.

### 3.3. Loss of the Potential Male Microbiota Protective Phenotype Is Associated with Genus and Species-Level Changes in Male BWF1 Microbiota

Our previous observations presented us with the opportunity to explore whether and which sex-based differences in commensal abundance at the genus/species level are associated with protection from disease. We first analyzed female and male BWF1 microbiota down to the genus and species levels in feces collected shortly before the move from the A Tower (2016) and in the months following the move to the CTRB (CTRB Early-2017–2018). In the A Tower, *Bacteroides* abundance, in general, was high (~20–30%), and significantly higher in adult male BWF1 mice than in adult female BWF1 mice (Figure 7A). However, this was not the case in the CTRB Early experiments. In three of the CTRB Early experiments, *Bacteroides* levels were almost undetectable in both female and male BWF1 mice (Figure 8A and Appendix A). Moreover, the levels of *Clostridium* abundance in the CTRB Early experiments (~40–60%; Figure 8A and Appendix A) were three times greater than was seen in the A Tower (~15–20%; Figure 7A). In a fourth CTRB Early experiment, *Bacteroides* was present, but was actually higher in female than male BWF1 mice, and *Clostridium* was still considerably higher than had been seen in the A Tower (Appendix A). The average *Bacteroides* abundance levels in males during the CTRB early period were 9.4 ± 3.5 (Mean + SEM) vs. 29 ± 7.0 during the A Tower period, and the average *Clostridium* abundance levels in males during the CTRB Early period were 43 ± 2.3 vs. 15.8 ± 3.8 during the A Tower period. The microbiota analysis also identified another genus, *Alistipes* (in particular, *Alistipes putredinis* and *Alistipes timonensis* species), that was present in the A Tower period (Figure 7), but was undetectable in the CTRB Early period (Figure 8 and Appendix A). Species analysis of the microbiota showed that *Bacteroides acidifaciens* and *Clostridium leptum* were the predominant *Bacteroides* and *Clostridium* species, respectively. These species, when present, exhibited the same patterns as their genera in both the A Tower and CTRB Early experiments (Figure 7B and Figure 8B and Appendix A). Our findings showed that a dramatic and destabilizing impact on the composition of the microbiota existed when transfer protection was lost.

### 3.4. Restoration of Potential Male Microbiota Protective Phenotype Is Associated with High Levels of Bacteroides and Low Levels of Clostridium

We next turned to the in-depth analysis of the microbiota in the feces collected from mice during the CTRB Recent period (2019–present) when the protective transfer phenotype (Figure 5) was reestablished in the CTRB. We found that the CTRB Recent period coincided with another shift of the microbiota composition to one that more resembled that seen in the A Tower. In two experiments (Figure 9 and Appendix A) conducted during the CTRB Recent period, *Bacteroides* (Figure 9A and Appendix A), particularly the *Bacteroides acidifaciens* species (Figure 9B and Appendix A), is once again present and at high levels, and was found at consistently greater abundance in male compared to female BWF1 mice. Also, the levels of *Clostridium* (Figure 9A and Appendix A), again particularly *Clostridium leptum* (Figure 9B and Appendix A), were back down to levels at or even below (~5–10%) those seen in the A Tower in both female and male BWF1 mice compared to the considerably elevated levels (40–60%) found during the CTRB Early period (Figure 8 and Appendix A). Additionally, *Alistipes*, which had been present in the A Tower (Figure 7), but undetectable in the CTRB Early experiments (Figure 8 and Appendix A), was detected in all of the CTRB Recent experiments (Figure 9 and Appendix A). The same species of *Alistipes* that were predominant in the A Tower, *Alistipes putredinis* and *Alistipes timonensis*, were also predominant in the CTRB Recent experiments (Figure 7B, and Figure 9B and Appendix A, respectively). The shift in microbiota profiles that coincided with the change in our mouse facility from the A Tower to the CTRB had reverted during the CTRB recent period to be more similar to the profiles seen in the A Tower [see Appendix A for a comparison of *Bacteroides* and *Clostridium* abundances between time periods; the average *Bacteroides* abundance levels in males during the A Tower period were 29 ± 7.0 (Mean ± SEM), for the CTRB Early period they were 9.4 ± 3.5, and for the CTRB Recent period were 35.1 ± 4.6, whereas the average *Clostridium* abundance levels in males during the A Tower period were 15.8 ± 3.8, for the CTRB Early period they were 43 ± 2.3, and for the CTRB Recent period they were 9.9 ± 1.7].

### 3.5. Protection from Disease by Male-to-Female Microbiota Transfer Correlates with Elevated Bacteroides and Decreased Clostridium Abundances in the Recipients

To confirm which gut microbiota genus/species are involved in the protection of lupus-prone mice seen after male microbiota transfer, we performed an in-depth analysis of microbiota data from microbiota transfer experiments conducted during the CTRB Early period (when transfer was not protective) and the CTRB Recent period (when transfer-mediated protection was reestablished). In the CTRB Early cecal transfer experiment, where male microbiota failed to protect (Figure 3), *Bacteroides* levels (which were predominantly the *Bacteroides acidifaciens* species) were high in all of the groups, but they were lower in female recipients of male microbiota compared to recipients of female microbiota (Figure 10). However, like the microbiota profiles of female and male mice in the CTRB Early experiments described above (Figure 8), all groups had very high levels of *Clostridium* (30–40%), and undetectable levels of *Alistipes* (Figure 10). In contrast, analysis of a microbiota transfer experiment from the CTRB Recent period found that recipients of male microbiota, when male transfer protected, had significantly higher *Bacteroides* abundances compared to the female BWF1 recipients of female microbiota (Figure 11). Importantly, the levels of *Clostridium* were considerably lower for all groups (10–20%) in the CTRB Recent transfer experiment (Figure 11) in comparison to the levels found in CTRB Early transfer experiment (30–40%; Figure 10 and Appendix A). Furthermore, unlike during the CTRB Early period when *Alistipes* was completely absent (Figure 10), *Alistipes* was detected in all recipients during the CTRB Recent period (Figure 11A), and the same species (Figure 11B) were predominant as in the A Tower (Figure 7B) female and male mice (*Alistipes timonensis* and *Alistipes putredinis*). This microbiota profile in the CTRB Recent cecal transfer experiment was highly stable, as we tested the microbiota composition again 4 weeks later (Figure 11C) and found the same higher levels of *Bacteroides* in the recipients of male microbiota, as well as lower levels of *Clostridium* and detectable *Alistipes* in all groups (Figure 11A,C).

Additionally, species-level analysis of the CTRB Early and CTRB Recent transfer experiments showed that as in the other experiments, *Bacteroides acidifaciens* and *Clostridium leptum* comprised almost all of the *Bacteroides* and *Clostridium* present, respectively (Figure 10B and Figure 11D). To address the possibility that high levels of *Clostridium* might interfere with the disease-suppressing activities of other bacteria, we calculated the *Bacteroides/Clostridium* ratio, and compared the CTRB Early and Recent transfer experiments. We found the *Bacteroides/Clostridium* ratio differed little in the female recipients of female microbiota between CTRB Early and CTRB Recent periods, but in recipients of male microbiota, the *Bacteroides/Clostridium* ratio was much higher in the CTRB Recent transfer experiment 4 weeks post-transfer at both 15 and 19 weeks of age (Figure 12) compared to the CTRB Early period. The *Bacteroides/Clostridium* ratio was also higher in males than females, in general, during the A Tower and CTRB Recent periods, but not during the CTRB Early period (Appendix A). Taken together, these data indicate that disease suppression mediated by male microbiota correlated with a high *Bacteroides/Clostridium* ratio that generally reflected high *Bacteroides* and low *Clostridium* abundance levels.

## 4. Discussion

The gut microbiota is critical for the normal development of the systemic as well as the mucosal immune systems and the influence is bi-directional, since the immune system dramatically influences the gut microbiota [4,41,45,46]. Consequently, it is not surprising that the gut microbiota very likely plays a significant role in the development and/or pathogenicity of a variety of diseases including autoimmune diseases [1,2,3,4,47,48]. Alterations in microbiota, sometimes referred to as dysbiosis, have been associated with a variety of diseases, including many autoimmune diseases such as systemic lupus erythematosus (SLE) [9,20,21]. The cause-and-effect relationship between microbiota and development/pathogenesis of the diseases is challenging to investigate in humans because the microbiota composition is evaluated only after disease is diagnosed, making it difficult to determine if the alteration in microbiota is a “cause” of or a contributor to disease, or is a consequence of the disease process (e.g., inflammation, etc.). For this reason, mouse models of disease can provide invaluable information because the microbiota can be evaluated before disease onset and followed through disease development and progression. A further complication affecting the study of the gut microbiota in humans is the significant impact environmental (non-genetic) factors have on the microbiota composition and function, including food, water, geographic area, etc.; all factors that are very difficult to control for.

This challenge is underscored by our findings in the current study with small changes in our mouse facility. Simply moving to a new barrier facility with all other factors remaining the same (i.e., food, water source, bedding, housing, husbandry protocols, environmental factors such as temperature/humidity/air quality, etc.) can have dramatic effects on the gut microbiota and its function, in this case, the ability of male microbiota to suppress disease after transfer to female recipients. In the current study, we initially took advantage of the differences in disease susceptibility between female (lupus-susceptible) and male (lupus-resistant) BWF1 mice to evaluate the potential for therapy involving fecal transplants. The challenges we experienced with these experiments ultimately led us to the serendipitous discovery of bacterial populations that may actually collaborate to mediate or obstruct protection from lupus. In the current study, we initially found that the composition of the gut microbiota of adult female BWF1 mice prior to disease differs significantly from the composition of age-matched male BWF1 mice. This showed that in addition to sex-based differences in disease, there were also sex-based differences in the microbiota in adult BWF1 mice, and that lupus-susceptible female BWF1 mice have divergent microbiota profiles from lupus-resistant male BWF1 mice. Since male mice did not develop disease, we wondered whether their gut microbiota could protect female mice from disease, and proceeded to conduct cecal transplant experiments. We found that, indeed, male microbiota fed to female mice prior to disease could significantly delay disease as well as decrease disease severity and increase survival. This experiment was conducted multiple times with similar success in our initial barrier facility (A Tower period). However, when we moved our mouse colony to a new barrier facility (in the CTRB), we lost that phenotype (male microbiota protection in female recipients) for two years despite making a concerted effort to ensure that the animal husbandry protocols and housing remained the same—e.g., food, water source, bedding, etc.—between the two facilities. This loss of protective phenotype by the male microbiota was associated with a dramatic change in gut microbiota composition in the new facility (CTRB) compared to the microbiota composition in the initial facility (A Tower). The change in microbiota composition in the CTRB had no effect on disease incidence, severity, or survival in either female or male BWF1 mice directly, but rendered the male cecal transplants ineffective in suppressing disease in the female recipients (referred to as the CTRB Early period). During the CTRB Early period, unlike the A Tower period, there was very little difference in microbiota composition between the female and male mice. In an effort to recover the protective phenotype, we made some changes to the animal care, including a change in animal room within the same barrier, water source (from filtered autoclaved tap water to autoclaved RO water), and a small change to the disinfecting protocol. Although we cannot conclude with any certainty which, if any, of these changes had an effect, we have found that since these changes were implemented, transfers of male microbiota again consistently suppress disease in female BWF1 recipients (referred to as the CTRB Recent period). Analyses of the microbiota from female and male mice during the A Tower period (when the cecal transplants protected), the CTRB Early period (when the cecal transplants did not protect), and the CTRB Recent period (when the cecal transplants once again protected) revealed that the microbiota composition from male BWF1 mice from the CTRB Recent period much more closely resembled the microbiota composition from the A Tower than the CTRB Early period. In particular, the abundance of *Bacteroides* (*Bacteroides acidifaciens* species) was high in male mice, whereas the abundance of *Clostridium* (*Clostridium leptum* species) was low in both female and male mice during both the A Tower and CTRB Recent periods. Furthermore, *Alistipes* (*Alistipes putredinis* and *Alistipes timonensis* species), which had been present in the A Tower and was completely missing in the CTRB Early period, re-emerged during the CTRB Recent period. When we compared the microbiota composition in the cecal content recipients during the CTRB Early period vs. the CTRB Recent period, we found the same patterns: increased *Bacteroides*, particularly the species *Bacteroides acidifaciens*, in the recipients of male microbiota compared to recipients of female microbiota, decreased *Clostridium* (particularly, the species *Clostridium leptum),* and the presence of *Alistipes* (in particular, the species *Alistipes putredinis* and *Alistipes timonensis*) during the CTRB Recent period (when the cecal transplants protected), by comparison to cecal contents recipients during the CTRB Early period (when the cecal transplants did not protect). In addition, the wildly disparate levels of *Bacteroides* abundance between CTRB Early Experiments 1–3 (Figure 8A and Appendix A) and CTRB Early Experiment 4 (Appendix A) that took place later during the CTRB Early period could indicate that the microbiota composition of the mice was in flux, although it should be noted that the *Clostridium* levels remained very high in all four CTRB Early experiments (Figure 8A and Appendix A).

The fact that male BWF1 microbiota can be protective suggests that the bacteria that are more abundant in male BWF1 mice may be acting to suppress disease. The only taxa consistently enriched in male BWF1 mice during the A Tower and CTRB Recent periods, periods during which the male microbiota protected against disease (Figure 2 and Figure 5, respectively), were the genus *Bacteroides* and, in particular, the species *Bacteroides acidifaciens* (Figure 7 and Figure 9, respectively). This makes *Bacteroides* a possible candidate for being a mediator of male microbiota disease suppression. However, if increased *Bacteroides* does promote protection against disease, then suppression of disease in female recipients of male microbiota should correlate with the presence of high levels of *Bacteroides* (received via transfer of cecal contents). In addition, the fact that higher levels of *Clostridium* correlate with a failure of the male microbiota to protect against disease raises the possibility that an overabundance of *Clostridium* might interfere with the disease-suppressing activities of other bacteria, and that the ratio of *Bacteroides* to *Clostridium* may be critical to identify a protective phenotype.

The higher abundance of *Bacteroides* in our model system is similar to what has been found in studies on autoimmune rheumatoid arthritis in humans. Multiple studies have shown that healthy controls have significantly higher abundance of *Bacteroides* than rheumatoid arthritis patients [49,50]. Furthermore, *Bacteroides acidifaciens* in particular has been associated with anti-inflammatory phenotypes. A previous study found that in the non-obese diabetes (NOD) mouse model, a therapeutic diet increased both the survival rate and *B*. *acidifaciens* abundance. The increase in *B*. *acidifaciens* correlated with changes in a variety of immune parameters that shifted systemic immunity towards a more anti-inflammatory profile [51]. Similarly, in a study examining the colitis-suppressing activities of a *Lactobacillus* probiotic, it was found that treatment with *Lactobacillus* resulted in a massive increase in *B*. *acidifaciens* abundance, and this correlated with suppression of colitic inflammation and restoration of gut homeostasis [52]. Anti-inflammatory properties of *Bacteroides acidifaciens* have also been found to alter inflammation in liver disease as well as dextran sodium sulfate (DSS)-induced colitis [53,54]. Another study found that administering *B*. *acidifaciens* to mice fed a high-fat diet protected them against obesity and the development of obesity-derived inflammation [55]. Taken together, these data support the possibility that increased abundance of *B*. *acidifaciens* could suppress the inflammatory immune responses seen in lupus.

On the other hand, our data suggest that the high levels of *Clostridium* may promote lupus progression and/or block any beneficial effects of high *Bacteroides* levels. There is some evidence in the literature supporting this possibility. A longitudinal analysis of SLE patient gut microbiota found that *C*. *leptum* was enriched in SLE patient gut microbiota compared to healthy controls, and that after treatment that successfully reduced disease activity, *C*. *leptum* abundance was decreased [56]. Furthermore, in another study, bacteria from the class Clostridia, which includes *Clostridium*, actively suppressed retinoic acid synthesis in the intestine [57]. Retinoic acid is important for the induction of tolerance-promoting T regulatory cells (Tregs) [58] and SLE is associated with lower levels of Tregs [59]. This suggests that an overabundance of *Clostridia* could potentially increase susceptibility to lupus by suppressing retinoic acid-dependent Treg induction.

A comparison of the *Bacteroides/Clostridium* ratios in our study suggested that the combination of high *Bacteroides* with low *Clostridium* may be key to the protective effect of male microbiota. Furthermore, *Alistipes* was absent when male cecal transfer failed to suppress disease in female mice and present when male cecal transfer suppressed disease in female mice, supporting the idea that *Alistipes* and *Bacteroides* could potentially be acting in concert. Findings like this could not only potentially lead to the development of new lupus therapies, but could also inform the use of current treatments. For example, the current frontline treatment for lupus is corticosteroids, but a recent study comparing SLE patients who had and had not undergone glucocorticoid treatment found glucocorticoid treatment was associated with decreased *Bacteroides* abundance [60]. If further research shows that higher *Bacteroides* abundance is beneficial for human SLE patients, the effects of a therapy on the microbiota might need to be considered when designing a treatment plan.

Fecal transplantation/microbiota transmission studies have been conducted in other mouse models of lupus with varying results. Johnson et al. have found that male microbiota fecal transplants into female SWRxNZBF1 (SNF1) mice delays lupus nephritis and decreases severity to some extent, but did not evaluate survival [33]. They analyzed gut microbiota at the genus level and found *Bacteroides* and *Parabacteroides* abundances were lower in male compared to female mice, whereas *Dysgonomonas* abundance was higher in female mice. In our study, we found the opposite for the *Bacteroides* genus, which was higher in male BWF1 mice during the two periods (A Tower and CTRB Recent) when the male microbiota was protective. We did not find detectable levels of either *Parabacteroides* or *Dysgonomonas* in either our female or male mice. These differences could be due to differences in mouse strains (i.e., genetic differences) and/or animal facility environments.

Taken together, our results emphasize that no single taxon can be considered in isolation, and that examination of the entire microbiological profile is essential for studies of the microbiota. For our data, male microbiota suppression of disease does not track with any one taxon, but with a combination of high *Bacteroides*, low *Clostridium,* and the presence of *Alistipes*. To the best of our knowledge, our study is the first to show that transfer of microbiota from lupus-resistant male mice can suppress disease and significantly delay mortality in lupus-susceptible female mice. Studies are currently ongoing to investigate the mechanisms mediating the suppression of disease via male cecal transfer. The results of microbiota studies in mice can be difficult to translate directly into human therapies for a number of reasons: the composition of the gut microbiota (as well as the host responses) can differ significantly between species (i.e., between mice and humans) and laboratory mice are maintained in a strictly controlled environment (e.g., diet, medications, pathogens, etc.) by comparison to human populations. However, this study provides evidence that some bacterial populations (or their products) possibly in collaboration with other gut bacteria can suppress disease and decrease mortality in mice. On the other hand, other bacterial populations, while not directly involved in causing disease, could potentially interfere with the protective activity of the disease suppressing bacterial populations. While the specific bacterial populations that have these respective activities may not be directly translatable to human therapies, understanding the underlying mechanisms by which they work may be very helpful in the development of therapies in humans. This study is the first step in that goal, and experiments are currently underway which are designed to understand the mechanisms underlying the protection mediated by these microbiota transplants, including the identification of metabolites produced by bacteria in the male gut microbiota that are capable of influencing the immune system, and subsequently, disease [13,34]. Future studies should also evaluate the virome and mycobiome as they may play an important role and have been found to be altered in lupus [61].

## Figures and Tables

**Figure 1 microorganisms-13-01023-f001:**
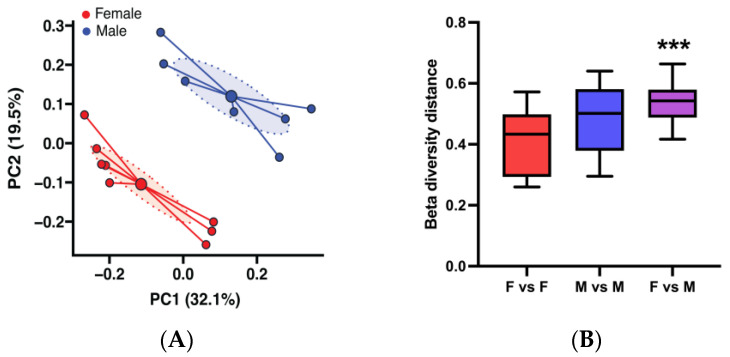
Adult female and male BWF1 mice have significantly different microbiota compositions. Feces were collected from 16-week-old female (F) and male (M) BWF1 mice during the A Tower period (2014–2016) and bacterial DNA was extracted. The 9 hypervariable regions of the 16S rRNA gene were sequenced and used to determine microbiota taxonomic composition. (**A**) Principal coordinate analysis (PCoA) plot of female and male BWF1 microbiota beta diversity using the Bray–Curtis dissimilarity metric. Small solid circles represent individual fecal samples, large solid circles represent the centroid, while ellipses represent 95% confidence intervals around the centroid of each group. (**B**) Female and male microbiota beta diversity distances were compared with the permutational multivariate analysis of variance (PERMANOVA) pairwise statistical test (*n* = 7). (*** *p* < 0.001).

**Figure 2 microorganisms-13-01023-f002:**
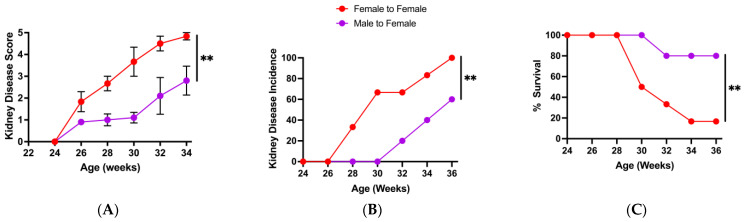
Transfer of male microbiota significantly suppresses kidney disease and enhances survival in female BWF1 mice. Female BWF1 mice were fed cecal contents (i.e., microbiota) via gavage from 16-week-old female or male BWF1 mice. Female-to-female (F-F, *n* = 6) and male-to-female (M-F, *n* = 5) cecal transfer recipients were monitored for kidney disease (glomerulonephritis) biweekly by measuring proteinuria in the urine. Proteinuria was scored on a scale of 1–5, with a score of 5 indicating death. Experiments were conducted during the A Tower period (2014–2016). (**A**) Kidney disease score. (**B**) Incidence of kidney disease (% mice with a proteinuria score of ≥3 for two consecutive biweekly readings). (**C**) Survival curve. Kidney disease scores were compared with two-way ANOVAs, and incidence and survival curves were compared using the Log rank (Mantel–Cox) test. ** *p* < 0.01.

**Figure 3 microorganisms-13-01023-f003:**
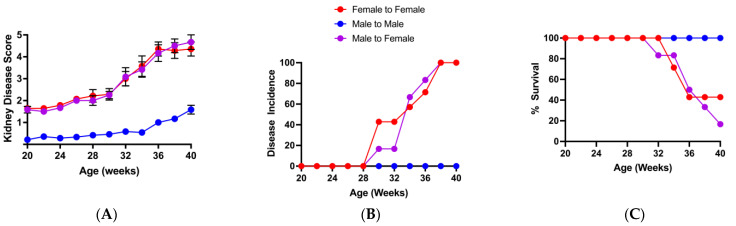
Transfer of male microbiota did not suppress kidney disease or enhance survival in female BWF1 mice during the CRTB Early period. Female BWF1 mice were fed cecal contents (i.e., microbiota) via gavage from 16-week-old female or male BWF1 mice. Female-to-female (F-F) and male-to-female (M-F) cecal transfer recipients were monitored for kidney disease (glomerulonephritis) biweekly by measuring proteinuria in the urine. Proteinuria was scored on a scale of 1–5, with a score of 5 indicating death (*n* = 6). Experiments were conducted during the CTRB Early period (2017–2018). (**A**) Kidney disease score. (**B**) Incidence of kidney disease (% mice with a proteinuria score of ≥3 for two consecutive biweekly readings). (**C**) Survival curve. Kidney disease scores were compared with two-way ANOVA, and incidence and survival curves were compared using the Log rank (Mantel–Cox) test.

**Figure 4 microorganisms-13-01023-f004:**
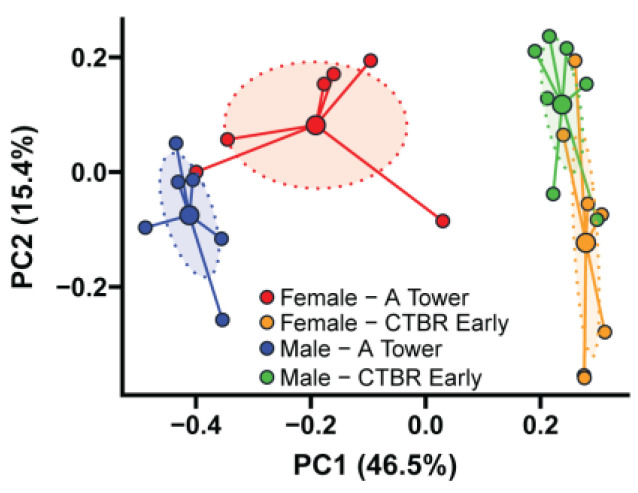
Female and male microbiota compositions differ between the A Tower and CTRB Early periods. Feces were collected from 16-week-old female and male BWF1 mice during either the A Tower (2014–2016; *n* = 6) or the CTRB Early (2017–2018; *n* = 7) periods, and bacterial DNA was extracted. The 9 hypervariable regions of the 16S rRNA gene were sequenced and used to determine microbiota taxonomic composition. Principal coordinate analysis (PCoA) plot of female and male BWF1 microbiota beta diversity using the Bray–Curtis dissimilarity metric. Small solid circles represent individual fecal samples, large solid circles represent the centroid while ellipses represent 95% confidence intervals around the centroid of each group.

**Figure 5 microorganisms-13-01023-f005:**
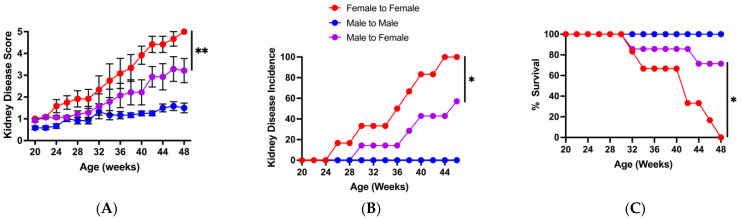
Transfer of male microbiota suppressed kidney disease and enhanced survival in female BWF1 mice during the CRTB Recent period. Female and male BWF1 mice were fed cecal contents (i.e., microbiota) via gavage from 16-week-old female or male BWF1 mice. Female BWF1 mice were fed cecal contents from adult male or female BWF1 mice. Female-to-female (F-F), male-to-female (M-F), and male-to-male (M-M) cecal transfer recipients were monitored for kidney disease biweekly by measuring proteinuria in the urine on a scale of 1–5, with a score of 5 indicating death (*n* = 6). Experiments were conducted during the CTRB Recent period (2019–present). (**A**) Kidney disease score. (**B**) Incidence of kidney disease (% mice with a proteinuria score of ≥3 for two consecutive biweekly readings). (**C**) Survival curve. Kidney disease scores were compared with two-way ANOVAs, and incidence and survival curves were compared using the Log rank (Mantel–Cox) test. * *p* < 0.05, ** *p* < 0.01.

**Figure 6 microorganisms-13-01023-f006:**
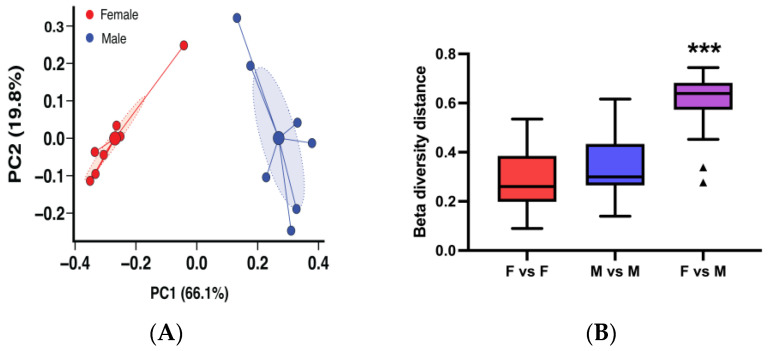
Female and male BWF1 mice have highly significantly different microbiota compositions during the CTRB Recent period. Feces were collected from 16-week-old female and male BWF1 mice during the CTRB Early (2019–present) period and bacterial DNA was extracted. The 9 hypervariable regions of the 16S rRNA gene were sequenced and used to determine microbiota taxonomic composition (*n* = 7). (**A**) Principal coordinate analysis (PCoA) plot of adult male and female BWF1 fecal microbiota beta diversity using the Bray–Curtis dissimilarity metric. Small solid circles represent individual fecal samples, large solid circles represent the centroid, while ellipses represent 95% confidence intervals around the centroid of each group. (**B**) Female and male microbiota beta diversity distances were compared with the permutational multivariate analysis of variance (PERMANOVA) pairwise statistical test. Outliers are indicated as triangles. (*** *p* < 0.001).

**Figure 7 microorganisms-13-01023-f007:**
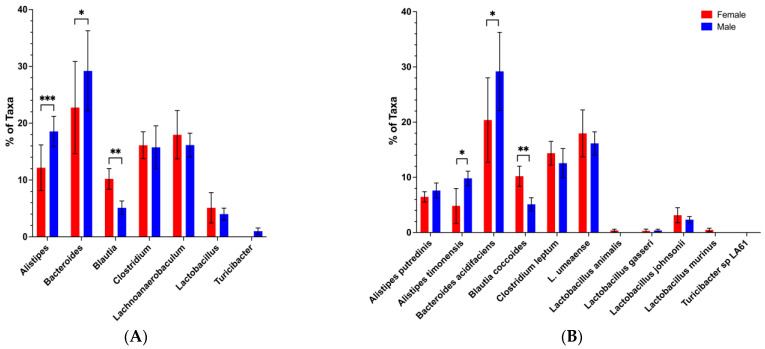
Abundance of *Bacteroides* was higher in adult male than female BWF1 mice during the A Tower period. Feces were collected from 16-week-old female and male BWF1 mice during the A Tower period (2016) and bacterial DNA was extracted. The 9 hypervariable regions of the 16S rRNA gene were sequenced and used to determine microbiota taxonomic composition (*n* = 7). Abundances were compared with One-way ANOVAs. (**A**) Genus-level microbiota from the A Tower–Female vs. Male experiment genus (*n* = 6). (**B**) Species-level microbiota from the A Tower–Female vs. Male experiment species (*n* = 6). * *p* < 0.05, ** *p* < 0.01, *** *p* < 0.001.

**Figure 8 microorganisms-13-01023-f008:**
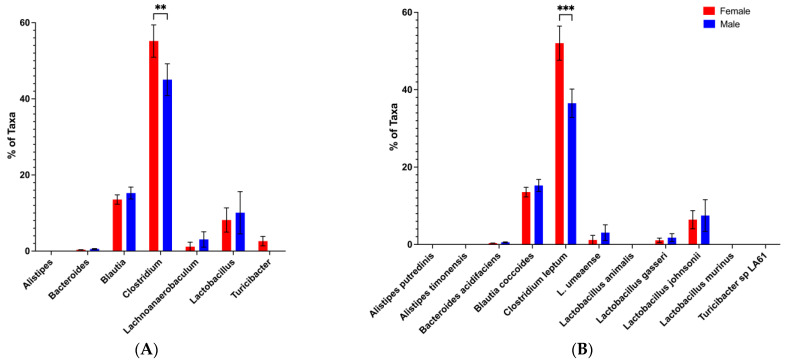
*Bacteroides* is, generally, very low, and abundance of *Clostridium* is very high in both females and males in the CTRB Early period. Feces were collected from 16-week-old female and male BWF1 mice during the CTRB Early (2017–2018) period and bacterial DNA was extracted. The 9 hypervariable regions of the 16S rRNA gene were sequenced and used to determine microbiota taxonomic composition. Abundances were compared with One-way ANOVAs. (**A**) Genus-level microbiota from Experiment 1 (*n* = 8). (**B**) Species-level microbiota from Experiment 1 (*n* = 8). ** *p* < 0.01, *** *p* < 0.001.

**Figure 9 microorganisms-13-01023-f009:**
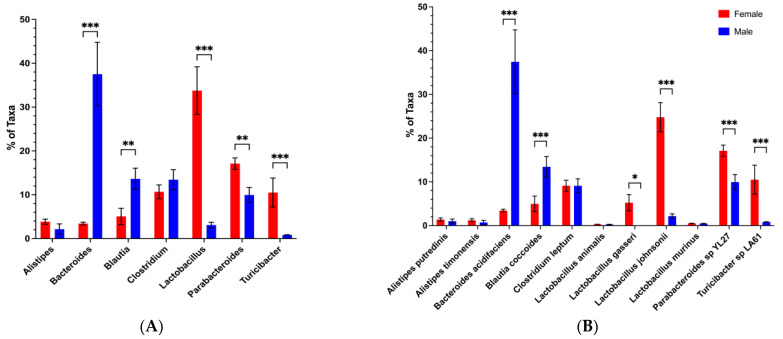
Abundance of *Bacteroides* was considerably higher in adult male than female BWF1 mice, and abundance of *Clostridium* levels was low in the CTRB Recent period. Feces were collected from 16-week-old female and male BWF1 mice during the CTRB Recent (2019–present) period and bacterial DNA was extracted. The 9 hypervariable regions of the 16S rRNA gene were sequenced and used to determine microbiota taxonomic composition. Abundances were compared with One-way ANOVAs. (**A**) Genus-level microbiota; (**B**) species-level microbiota. * *p* < 0.05, ** *p* < 0.01, *** *p* < 0.001.

**Figure 10 microorganisms-13-01023-f010:**
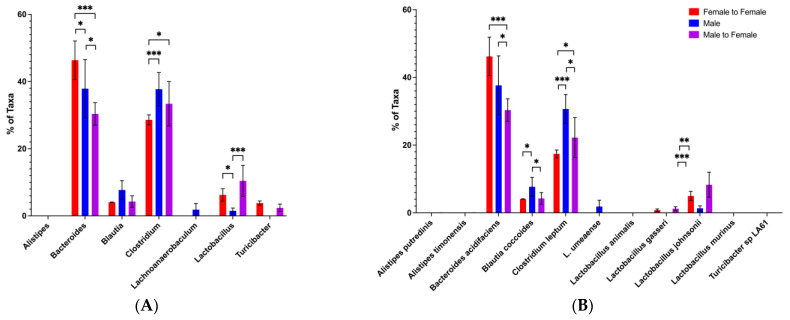
In CTRB Early microbiota transfer experiments, *Bacteroides* was present but abundance was lower in the male-to-female transfers and *Clostridium* was high in all experimental groups. During the CTRB Early (2017–2018) period, female BWF1 mice were fed cecal contents (i.e., microbiota) from 16-week-old female (Female-to-Female) or male (Male-to-Female) BWF1 mice via gavage monthly. Feces were collected from 16-week-old cecal transfer recipients (4 weeks after cecal transfer), and bacterial DNA was extracted. The 9 hypervariable regions of the 16S rRNA gene were sequenced and used to determine microbiota taxonomic composition. Abundances were compared with One-way ANOVAs (*n* = 6). (**A**) Genus-level microbiota; (**B**) species-level microbiota. * *p* < 0.05, ** *p* < 0.01, *** *p* < 0.001.

**Figure 11 microorganisms-13-01023-f011:**
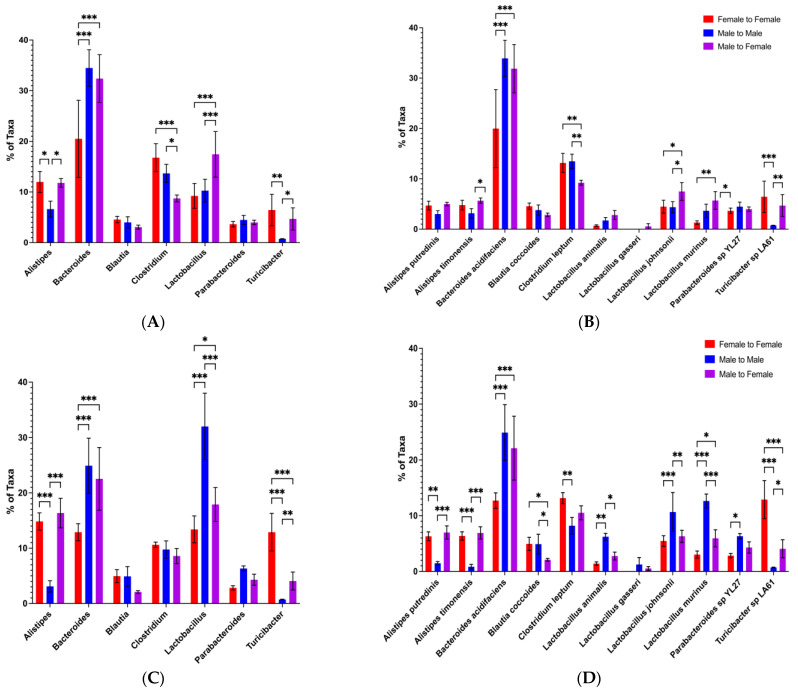
In CTRB Recent microbiota transfer experiments, abundance of *Bacteroides* was higher in the recipients of male microbiota, and abundance of *Clostridium* was low at two different time-points. During the CTRB Recent (2019–present) period, female BWF1 mice were fed cecal contents (i.e., microbiota) from 16-week-old female (female-to-female) or male (male-to-female) BWF1 mice via gavage monthly. Male mice (male-to-male) were fed male cecal contents as a control. Feces were collected 4 weeks after transfer and bacterial DNA was extracted. The 9 hypervariable regions of the 16S rRNA gene were sequenced and used to determine microbiota taxonomic composition. Abundances were compared with One-way ANOVAs (*n* = 6). (**A**) Genus-level microbiota in feces collected at 15 weeks of age; (**B**) species-level microbiota in feces collected at 15 weeks of age; (**C**) genus-level microbiota in feces collected at 19 weeks of age; (**D**) species-level microbiota in feces collected at 19 weeks of age. * *p* < 0.05, ** *p* < 0.01, *** *p* < 0.001.

**Figure 12 microorganisms-13-01023-f012:**
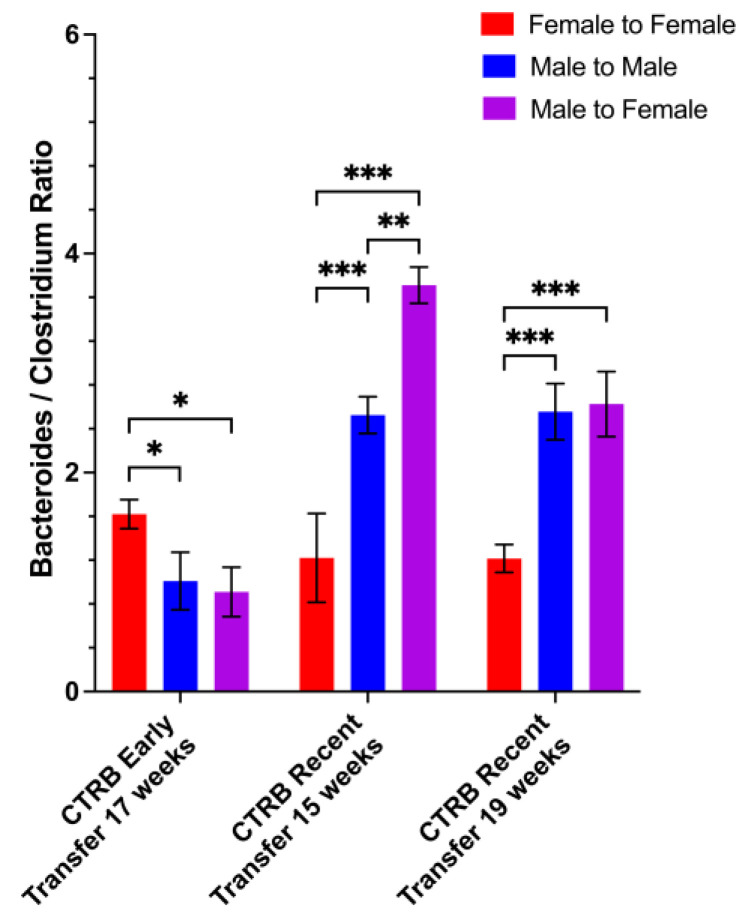
Ratios between *Bacteroides* and *Clostridium* abundance in cecal transfer experiments. Fecal samples from cecal transfer recipients were collected during the CTRB Early (2017–2018) and CTRB Recent (2019–present) periods and bacterial DNA was extracted. The 9 hypervariable regions of the 16S rRNA gene were sequenced and used to determine microbiota taxonomic composition. The *Bacteroides*/*Clostridium* ratios were calculated from the data shown in Figure 10 and Figure 11 by dividing average *Bacteroides* abundance by average *Clostridium* abundance, and are shown as *Bacteroides/Clostridium* ratios for the cecal recipient comparisons. * *p* < 0.05, ** *p* < 0.01, *** *p* < 0.001.

## Data Availability

The original contributions presented in this study are included in the article/Appendix A. Further inquiries can be directed to the corresponding author.

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
