# Peer review of "Characterization of Sex-Based Differences in Gut Microbiota That Correlate with Suppression of Lupus in Female BWF1 Mice"

_microorganisms, 2025, doi:10.3390/microorganisms13051023_

Round 1
Reviewer 1 Report
Comments and Suggestions for Authors
Characterization of Sex-Based Differences in Gut Microbiota That Correlate with Suppression of Lupus in Female BWF1 Mice.
Systemic lupus erythematosus (SLE) is more prevalent in females and is associated with microbiota dysbiosis in mice and humans. Gut microbiota composition was investigated in female and male NZBxNZWF1. The gut microbiota composition of BWF1 mice differed between adult male (disease-prone) and male (disease-resistant) mice. In male mice, Bacteroides abundance was high, and Clostridium was low during the periods when male cecal transplants suppressed disease. Similar patterns in were found in females receiving male cecal transplants, i.e., a high Bacteroides/Clostridium ratio. The researchers concluded that Bacteroides and Clostridium may have opposing effects on SLE suppression.
The abstract needs more information about the experiment, design, statistical analysis.
The introduction: L67-86: you need to provide references, distinguish between what was done by you in your lab and done by other researchers.
Experimental design is required, replication., etc.
Animal handling and the process of collecting fecal samples are missing.
Cecal contents from 16-week-old male or female BWF1 mice were collected, details are missing about the process of collecting ceca.
Results and discussion: excellent job, just formatting issues. Be consistent with providing italic for bacterial name
Author Response
Reviewer #1
Response: We would like to thank the reviewer for taking the time to provide a thoughtful review of our manuscript. We believe the resulting revisions will significantly strengthen the manuscript.
Systemic lupus erythematosus (SLE) is more prevalent in females and is associated with
microbiota dysbiosis in mice and humans. Gut microbiota composition was investigated in female and male NZBxNZWF1. The gut microbiota composition of BWF1 mice differed between adult male (disease-prone) and male (disease-resistant) mice. In male mice, Bacteroides abundance was high, and Clostridium was low during the periods when male cecal transplants suppressed disease. Similar patterns in were found in females receiving male cecal transplants, i.e., a high Bacteroides/Clostridium ratio. The researchers concluded that Bacteroides and Clostridium may have opposing effects on SLE suppression.
Comment: The abstract needs more information about the experiment, design, statistical analysis.
Response: The abstract has been extensively revised as requested. The word count for the abstract is limited to 200 by the journal, and we modified the abstract as best we could with this restriction.
Comment: The introduction: L67-86: you need to provide references, distinguish between what was done by you in your lab and done by other researchers.
Response (L75-85): This section of the Introduction has been extensively revised and what was a summary of our findings that are described in more detail in the results section has now been replaced by a description of experiments performed in response to another reviewer’s comments.
Comment: Experimental design is required, replication., etc.
Response: The subsections within Materials and Methods section have been extensively revised and include experimental design [specifically, Sections 2.1, 2.4, and 2.6 as well as information concerning replication (section 2.5, lines 186-189).
Comment: Animal handling and the process of collecting fecal samples are missing.
Response: A description of animal handling and collection of fecal samples has been added under the heading, 2.2. Animal Handling and Fecal Sample Collection.
Comment: Cecal contents from 16-week-old male or female BWF1 mice were collected, details are missing about the process of collecting ceca.
Response: A description of the collection and processing of the cecal contents has been added under the heading, 2.3. Cecal (microbiota) transplant preparation and processing.
Results and discussion: excellent job, just formatting issues. Be consistent with providing italic for bacterial name

Reviewer 2 Report
Comments and Suggestions for Authors
In the present article, the authors sought to evaluate the composition of the gut microbiota in relation to the sex of mouse models of systemic lupus erythematosus. The paper constitutes a novel approach to a pertinent issue, meticulously planned and executed with clarity. The following observations are offered with the intention of enhancing the content of the publication.
Specific comments:
Lines 5 and 6: It is recommended that the authors' country of origin be stated.
Line 67: It is advised that the presentation of research results obtained be avoided in the introduction. This should be incorporated into the conclusion.
Line 67 - The paper does not contain any hypothesis. It is imperative to remember that a hypothesis is not the purpose of the paper.
Line 103 - The ethics committee approval for the experiment (number, date of issue, etc.) is required.
Line 131 - It is unclear if the authors and how they tested the normal distribution.
Line 137 - What exactly two factors were considered in the two-way ANOVA?
Comments on the Quality of English LanguageThe English could be improved to more clearly express the research.
Author Response
Reviewer #2
Response: We would like to thank the reviewer for taking the time to provide a thoughtful review of our manuscript. We believe the resulting revisions will significantly strengthen the manuscript.
In the present article, the authors sought to evaluate the composition of the gut microbiota in relation to the sex of mouse models of systemic lupus erythematosus. The paper constitutes a novel approach to a pertinent issue, meticulously planned and executed with clarity.
The following observations are offered with the intention of enhancing the content of the publication.
Specific comments:
Comment: Lines 5 and 6: It is recommended that the authors' country of origin be stated.
Response: Completed as requested.
Line 67: It is advised that the presentation of research results obtained be avoided in the
introduction. This should be incorporated into the conclusion.
Response (L75-85): The introduction has been extensively modified to only include a brief description of the study while avoiding a presentation of the results.
Line 67 - The paper does not contain any hypothesis. It is imperative to remember that a hypothesis is not the purpose of the paper.
Response (L75-85): It unclear what the reviewer is referring in this comment. However, this section of the Introduction has been extensively revised which we hope addresses the comment.
Line 103 - The ethics committee approval for the experiment (number, date of issue, etc.) is required.
Response (L121-123): Protocol numbers and dates of approval by the University of Louisville Institutional Animal Care and Use Committee that span the period described from 2014 to present have been added to the manuscript.
Line 131 - It is unclear if the authors and how they tested the normal distribution.
Response: In this study, we used GraphPad Prism to assess the normality of the data. The data were normally distributed, and, therefore, parametric tests were applied (e.g., t-test or ANOVA). This has been described in Section 2.7 Statistical analyses.
Line 137 - What exactly two factors were considered in the two-way ANOVA?
Response (Lines 216-227): Please see section 2.6 for an extensive description of the factors considered.

Reviewer 3 Report
Comments and Suggestions for Authors
microorganisms-3490782-peer-review-v1
This paper is a very good example of how an interesting research project can be performed but then formatted and presented in a not very good way. Authors have well planned (well, with some administrative interruptions during performing the work) and performed the experimental work. The results obtained was interesting and gave some novelty regarding lupus. However, when the manuscript was prepared, it is clearly seen the missing of experience (I believe that this is the reason for the not satisfactory preparation of the manuscript, and not negligence of the authors). Thus, in my opinion manuscript need to be given a second chance and ask authors to reorganize the manuscript, provide the needed missing parts (several of them into the material and methods sections) and critically reorganize the Results and Discussion sections with appropriate focus on purpose of the mentioned sections. Moreover, 58% of similarities are very high. This needs attention from the authors
The title will be appropriate to change a bit. Authors in fact have some changes in the environment, and later in the manuscript discussed how this influenced the experimental procedures and obtained results. Will it be appropriate if this can be stated in the title.
In the affiliation, Ln7, Please provide details regarding to the authors affiliations. The journal template have a very well stated model of how this needs to be. Obvious, you know where University of Louisville is located, however, for other readers, this can be Canada or Mexico, or in other parts of the world. Moreover, a journal requires that e-mail for all authors can be provided. Simply follow the instructions form the journal. Thus, maybe help from more experience colleagues can always be option for formatting the manuscript.
The introduction is well prepared and provides well-structured information regarding further investigated topics in the research manuscript. Authors have focus on the existing knowledge regarding lupus and sex differences of development of the disease and have justified their research hypothesis and way to investigate this.
Ln104, Ln109, etc. Please, provide always names of the city, state and country. In fact, in the new world several times the same city name can be repeated several times. This is just for accuracy and to avoid misunderstanding..
In several places authors refer to extended list of references. This is a good point, and showing that authors have scanned the literature (well, 58% similarity), however, when it is possible to reduce some of the references used will be a good option. In some cases, referring to the review paper is better than providing 5-6 or even more research papers. Please, try to reduce references a bit.
In section 2.2. Please, check if the provided references are cited according to the requirements form the Publisher and Journal.
Regarding the procedure provided in section 2.3., do you have done this according to previously established procedure or was it your original idea to do it this way? Maybe a reference will be provided for the applied procedure. The same prepared suspension was used for all feeding, or each time a new suspension was prepared. If the suspension was the same, how was the suspension stored to ensure that microbial population was not altered?
In the results section, please, avoid not needed repetitions from the introduction. Try to be more objective and focus not on justification of the experiments, but on the observed facts. IN some cases, it is more appropriate if the results and discussion section can be combined, to avoid not needed repetitions, however, this needs to be agreed with the Editor of the Journal.
In the material and methods authors do not provide details for all experimental procedures. For example, how was the kidney disease score calculated? Please be sure that all performed experimental plans, presented as results and later discussed need to be well stated how was performed. In some cases, standard procedures from some areas seems clear, but clear for the authors, but not for readers of the manuscript.
Subsections used in the results section looks like summary of the obtained results. This is an interesting approach for presenting, however, maybe authors can be a bit more conservative and try to provide title/subtitle that give idea about the context of the following presented material and not conclusion of the observed results.
The results section will need to be reorganized and adjusted to the way that will present only observed facts and do not go to discussions. Moreover, discussion is quite a bit limited, but this is understander, since the Results section is over detailed and enriched with the discussion elements. Simple, remove all the discussion from the results section and move it to the Discussion section. Please avoid not needed repetitions and be more objective to the previously stated goals and objectives.
Please, consider asking for permission from the Editor of the journal and combine Results and Discussion, I think this will be the optimal way for the presentation of the current manuscript.
In the description of the figures, try to be more objective and avoid not needed discussion of the results.
Maybe help form more experience collogues will be a good option to be better formatting and presenting the manuscript.
References are not into the style of the Publisher and Journal.
Author Response
Reviewer #3
Response: We would like to thank the reviewer for taking the time to provide a thoughtful review of our manuscript. We believe the resulting revisions will significantly strengthen the manuscript.
This paper is a very good example of how an interesting research project can be performed but then formatted and presented in a not very good way. Authors have well planned (well, with some administrative interruptions during performing the work) and performed the experimental work. The results obtained was interesting and gave some novelty regarding lupus. However, when the manuscript was prepared, it is clearly seen the missing of experience (I believe that this is the reason for the not satisfactory preparation of the manuscript, and not negligence of the authors). Thus, in my opinion manuscript need to be given a second chance and ask authors to reorganize the manuscript, provide the needed missing parts (several of them into the material and methods sections) and critically reorganize the Results and Discussion sections with appropriate focus on purpose of the mentioned sections. Moreover, 58% of similarities are very high. This needs attention from the authors.
Comment: The title will be appropriate to change a bit. Authors in fact have some changes in the environment, and later in the manuscript discussed how this influenced the experimental procedures and obtained results. Will it be appropriate if this can be stated in the title.
Response: We apologize for the confusion. The paper focuses on the unexpected change in the ability of male microbiota to protect female recipients from disease due to a shift in microbiota composition and has allowed us to try to identify bacterial populations that play a role in protection. The paper does not focus on the change in environment influencing the microbiota (a generally accepted phenomenon) since we cannot characterize the factors responsible for that change. Because it is a retrospective study using archived samples and historical information, we do not know which if any of the factors had an impact on the microbiota or their function. For this reason, we believe that it is more accurate to maintain the focus on the differences in female and male microbiota and which differences may play a role in protection from disease rather that a change in environment that influences microbial populations. Some of the titles in the Results section have been revised to reflect that.
Comments: In the affiliation, Ln7, Please provide details regarding to the authors affiliations. The journal template have a very well stated model of how this needs to be. Obvious, you know where University of Louisville is located, however, for other readers, this can be Canada or Mexico, or in other parts of the world. Moreover, a journal requires that e-mail for all authors can be provided.
Response: The emails for each author have been added to the manuscript as suggested.
Simply follow the instructions form the journal. Thus, maybe help from more experience
colleagues can always be option for formatting the manuscript.
The introduction is well prepared and provides well-structured information regarding further investigated topics in the research manuscript. Authors have focus on the existing knowledge regarding lupus and sex differences of development of the disease and have justified their research hypothesis and way to investigate this.
Comments. Ln104, Ln109, etc. Please, provide always names of the city, state and country. In fact, in the new world several times the same city name can be repeated several times. This is just for accuracy and to avoid misunderstanding..
Response (L89-90): City, state and country have been added as suggested.
Comments. In several places authors refer to extended list of references. This is a good point, and showing that authors have scanned the literature (well, 58% similarity), however, when it is possible to reduce some of the references used will be a good option. In some cases, referring to the review paper is better than providing 5-6 or even more research papers. Please, try to reduce references a bit.
Response. As requested by the reviewer, we have decreased the number of references overall. However, we have opted to provide original research references instead of review papers since in our experience, citations and interpretations of papers can be inaccurate in review papers.
Comment. In section 2.2. Please, check if the provided references are cited according to the requirements form the Publisher and Journal.
Response. References have been modified to adhere to the requirements of the journal.
Comment: Regarding the procedure provided in section 2.3., do you have done this according to previously established procedure or was it your original idea to do it this way? Maybe a reference will be provided for the applied procedure. The same prepared suspension was used for all feeding, or each time a new suspension was prepared. If the suspension was the same, how was the suspension stored to ensure that microbial population was not altered?
Response: The Material and Methods section has been significantly modified to address these issues, and procedures are now described in extensive detail and references have been added in new Sections 2.3 Cecal (microbiota) transplant preparation and processing and 2.4 Cecal microbiota transfer.
In the results section, please, avoid not needed repetitions from the introduction. Try to be more objective and focus not on justification of the experiments, but on the observed facts. IN some cases, it is more appropriate if the results and discussion section can be combined, to avoid not needed repetitions, however, this needs to be agreed with the Editor of the Journal.
Response: The introduction has been extensively modified (L75-85) and results have been removed and replaced by a general description of the study design. “Discussion” has been removed from the Results when possible.
Comment: In the material and methods authors do not provide details for all experimental procedures. For example, how was the kidney disease score calculated?
Response: We apologize for the omission. A description of the kidney disease assessment is now in new section 2.5.
Please be sure that all performed experimental plans, presented as results and later discussed need to be well stated how was performed. In some cases, standard procedures from some areas seems clear, but clear for the authors, but not for readers of the manuscript.
Response: We have added experimental descriptions in various sections including Materials and Methods, etc in an attempt to provide better clarity.
Comments: Subsections used in the results section looks like summary of the obtained results. This is an interesting approach for presenting, however, maybe authors can be a bit more conservative and try to provide title/subtitle that give idea about the context of the following presented material and not conclusion of the observed results.
The results section will need to be reorganized and adjusted to the way that will present only observed facts and do not go to discussions. Moreover, discussion is quite a bit limited, but this is understander, since the Results section is over detailed and enriched with the discussion elements. Simple, remove all the discussion from the results section and move it to the Discussion section. Please avoid not needed repetitions and be more objective to the previously stated goals and objectives.
Please, consider asking for permission from the Editor of the journal and combine Results and Discussion, I think this will be the optimal way for the presentation of the current manuscript.
In the description of the figures, try to be more objective and avoid not needed discussion of the results.
Response: We modified the Results section and have tried to be more objective and to focus on the loss of protective phenotype (focus of the paper) rather than change in facilities. Most of the discussion in the results section has been removed. The titles for some of the Results sections have been modified to better reflect the content.
Maybe help form more experience collogues will be a good option to be better formatting and presenting the manuscript.
References are not into the style of the Publisher and Journal.
Response. References have been modified to adhere to the requirements of the journal.

Reviewer 4 Report
Comments and Suggestions for Authors
Overall, the manuscript presents an interesting and relevant topic, exploring sex-based differences in gut microbiota in mice in relation to SLE. However, several methodological flaws are present, and some key information appears unclear. The study design requires further attention, as important details are missing, such as the number of mice used, the groups, and the controls. Providing this information is essential to ensure the clarity and reproducibility of the study. Furthermore, the presentation of the results lacks clarity, making it difficult to discern which groups were analyzed. The comparisons between animal facilities, sex, and age groups further contribute to the confusion, and this section would benefit from more precise explanations and a more organized structure.
Additionally, while I have included some written feedback in my review, I strongly recommend a thorough revision for conciseness and fluency, as some sections are challenging to follow.
Lines 31- 33: I suggest modifying this statement: “Dysbiosis, or an alteration in the gut microbiota, is linked to factors such as antibiotic use, diet, infections, and underlying health conditions. It is also frequently associated with autoimmune diseases, including type 1 diabetes, multiple sclerosis, rheumatoid arthritis, and systemic lupus erythematosus (SLE), in both humans and mouse models”.
Line 34: I suggest modifying this statement: “Sex-based differences in the microbiota have been identified in both humans and mice, with variations in the composition and diversity of microbial communities between males and females.”
Line 36: Modify it: “In SLE, women comprise over 90% of patients”.
Line 42: Cite the “Systemic lupus erythematosus” in full once in the text, then you can use the abbreviation. Keep it standardized throughout the text.
Line 46: Modify it to: “Alterations in the gut microbiota have been associated with both SLE patients and mouse models”.
Lines 49 - 51: Modify it to: “Compared to healthy controls, higher abundance of Ruminococcus gnavus in SLE patients and elevated levels of antibodies against R. gnavus have been shown to correlate with overall antibody levels and kidney damage”.
Lines 51-53: Modify it to: “The presence of Enterococcus gallinarum in the liver of SLE patients has been correlated with higher levels of auto-antibodies, while the presence of Ruminococcus is associated with decreased Treg levels in SLE patients”.
Lines 69-86: I suggest removing the statements where the authors commented on some results and keeping only the paragraph with the study's aim.
Materials and Methods: I recommend restructuring this section, beginning with the "Animals" subsection, followed by "Cecal Microbiota Transfer." A more detailed explanation of the study design should also be provided for better understanding. “Microbiota Composition Analysis and Statistical Analysis” should be kept in one section.
Line 108: Provide the library kit and sequencing technology that was performed.
Line 124: I recommend providing more detailed information about the study design, as it currently appears incomplete. Specifically, describe the mouse groups evaluated in this study, including the experimental and control groups. It would also be helpful to include information on the number of doses administered, the duration of treatment, and details about animal harvest and sample collection for microbiome evaluation. What was the duration of this experiment? How many mice were used in this study?
Line 128: Replace cecal feeding with FMT (fecal microbiota transplantation).
Lines 147- 149: How did the authors evaluate anti-nuclear antibodies, as this method was not mentioned in the Materials and Methods section?
Line 200: The criteria for evaluating the mice's disease should be established and mentioned.
- The study shows that male mice in the CTRB Early period lost their protective microbiota, but the exact factor responsible remains unclear.
- Have you analyzed whether external environmental factors (e.g., air quality, humidity, diet batch differences) changed between facilities?
- Since the water source and disinfection protocol were modified, can you further isolate their effects by systematically testing each variable?
- The study primarily focuses on microbiota shifts, but how do these correlate with host immune responses?
- Have cytokine levels, immune cell populations, or other immune markers been analyzed to confirm that disease suppression is microbiota-driven rather than influenced by other physiological changes?
- Could a germ-free or antibiotic-treated mouse model, recolonized with protective microbiota, confirm causality?
- Investigating whether fecal microbiota transplantation (FMT) or specific bacterial consortia can replicate the protective effect could enhance translational potential.
A revision of the entire text is required.
Author Response
Reviewer #4
Response: We would like to thank the reviewer for taking the time to provide a thoughtful review of our manuscript. We believe the resulting revisions will significantly strengthen the manuscript.
Overall, the manuscript presents an interesting and relevant topic, exploring sex-based
differences in gut microbiota in mice in relation to SLE. However, several methodological flaws are present, and some key information appears unclear. The study design requires further attention, as important details are missing, such as the number of mice used, the groups, and the controls. Providing this information is essential to ensure the clarity and reproducibility of the study. Furthermore, the presentation of the results lacks clarity, making it difficult to discern which groups were analyzed. The comparisons between animal facilities, sex, and age groups further contribute to the confusion, and this section would benefit from more precise explanations and a more organized structure.
Additionally, while I have included some written feedback in my review, I strongly recommend a thorough revision for conciseness and fluency, as some sections are challenging to follow.
Comments. Lines 31- 33: I suggest modifying this statement: “Dysbiosis, or an alteration in the gut microbiota, is linked to factors such as antibiotic use, diet, infections, and underlying health conditions. It is also frequently associated with autoimmune diseases, including type 1 diabetes, multiple sclerosis, rheumatoid arthritis, and systemic lupus erythematosus (SLE), in both humans and mouse models”.
Response (L 38-41): Completed as requested.
Comments. Line 34: I suggest modifying this statement: “Sex-based differences in the microbiota have been identified in both humans and mice, with variations in the composition and diversity of microbial communities between males and females.”
Response (L 41-43): Completed as requested.
Line 36: Modify it: “In SLE, women comprise over 90% of patients”.
Response (L 45-46): Completed as requested.
Line 42: Cite the “Systemic lupus erythematosus” in full once in the text, then you can use the abbreviation. Keep it standardized throughout the text.
Response: Completed as requested.
Line 46: Modify it to: “Alterations in the gut microbiota have been associated with both SLE patients and mouse models”.
Response (L 56-57): Completed as requested.
Lines 49 - 51: Modify it to: “Compared to healthy controls, higher abundance of Ruminococcus gnavus in SLE patients and elevated levels of antibodies against R. gnavus have been shown to correlate with overall antibody levels and kidney damage”.
Response (L 59-61): Completed as requested.
Lines 51-53: Modify it to: “The presence of Enterococcus gallinarum in the liver of SLE patients has been correlated with higher levels of auto-antibodies, while the presence of Ruminococcus is associated with decreased Treg levels in SLE patients”.
Response (L 61-64): Completed as requested.
Lines 69-86: I suggest removing the statements where the authors commented on some results and keeping only the paragraph with the study's aim.
Response (L76-85): This section has been extensively revised in response to this and other reviewers' comments.
Comment: Materials and Methods: I recommend restructuring this section, beginning with the "Animals" subsection, followed by "Cecal Microbiota Transfer." A more detailed explanation of the study design should also be provided for better understanding. “Microbiota Composition Analysis and Statistical Analysis” should be kept in one section.
Response: The Methods and Materials section has been extensively revised in response to this and other reviewers’ comments.
Comment: Line 108: Provide the library kit and sequencing technology that was performed.
Response (L 203-207): Completed as requested.
Comments: Line 124: I recommend providing more detailed information about the study design, as it currently appears incomplete. Specifically, describe the mouse groups evaluated in this study, including the experimental and control groups. It would also be helpful to include information on the number of doses administered, the duration of treatment, and details about animal harvest and sample collection for microbiome evaluation. What was the duration of this experiment? How many mice were used in this study?
Response: The Methods and Materials section has been extensively revised in response to this and other reviewers’ comments. Section 2.4 Cecal microbiota transfer has been expanded to include the requested information.
Line 128: Replace cecal feeding with FMT (fecal microbiota transplantation).
Response: We apologize for the confusion but this is not fecal transplantation, it is cecal transplantation. It is different, and in papers using this mode of microbial transfer, it is referred to as cecal feeding (reference 31).
Lines 147- 149: How did the authors evaluate anti-nuclear antibodies, as this method was not mentioned in the Materials and Methods section?
Response: We apologize for the confusion, but anti-nuclear antibodies were not evaluated in this study. We evaluated proteinuria as a measure of kidney disease as described in new section 2.5.
Line 200: The criteria for evaluating the mice's disease should be established and mentioned.
Response: We apologize for the omission. The criteria for kidney disease is now described in new section 2.5.
Comment: The study shows that male mice in the CTRB Early period lost their protective microbiota, but the exact factor responsible remains unclear.
Comment: Have you analyzed whether external environmental factors (e.g., air quality, humidity, diet batch differences) changed between facilities?
Response: We had extensive discussions with critical Animal Facilities personnel prior to moving our animal colony to the new animal facility in January, 2017, and emphasized the need to maintain consistency between facilities. To the best of our knowledge, environmental factors that were under our and the animal facility personnel control were identical between both animal facilities, including food, water source, bedding, housing (including vent rack, caging, sterilizing methods), as well as humidity, temperature, light cycle, air quality. The change in protective phenotype after transfer of our animal colony to the new animal facility was unexpected. Because we breed the BWF1 mouse donors and recipients, and the animal model is spontaneous and female mice typically do not develop full-blown kidney disease until ~28 weeks of age, we were not even aware that the male microbiota transfers were no longer protective for more than 1 year after changing animal facilities. We repeated the transfer experiments multiple times over the two-year period with the same results, no protection. Near the end of the second year (late fall 2018), we made changes, some intentional (water source, small disinfecting change) and one that was out of our control (we were moved out of our animal room within the same facility to make way for construction of our new germ-free facility). We could not identify what had changed, since as far as we knew, nothing other than the building had changed. We were very focused on trying to restore the protective phenotype, and for this reason, we decided to improve the purity of our water source and changed it from our original 0.2 micron-filtered autoclaved tap water to reverse osmosis purified autoclaved water. The small disinfecting change was simply to request that our extremely careful animal care technicians to not directly touch the food in the cages with their gloves which had been freshly disinfected (we thought that perhaps disinfectant was getting into the food and altering the gut microbiota).
For this manuscript, we took advantage of the fact that we had collected feces from mice in both our original animal facility (A Tower period; when the cecal transfers protected) and in the new facility during the early period (CTRB Early period; when the cecal transfers did not protect). This allowed us to compare the microbiota in the feces from both of those facilities to the microbiota in the feces of mice collected during the later/more recent period in the new facility (CTRB Recent period). The goal of this study, therefore, was to use these comparisons to identify the bacterial populations in male microbiota that may play a role in protection from disease in female recipients. This study was not designed to determine the factor(s) that is(are) responsible for the loss in protective function by the male microbiota. We have now modified some of the titles in the results section to reflect that.
Comment: Since the water source and disinfection protocol were modified, can you further isolate their effects by systematically testing each variable?
Response: The water source was exactly the same between the original animal facility (A Tower period, when the protective protocol worked) and the early period in the new animal facility (CTRB Early, when the protective protocol stopped working), and therefore, it was not the water source that was responsible for the initial change in composition or protective function of male microbiota. The change in the disinfection protocol was very minor (as described above), and involved discontinuing brief handling of food with freshly disinfected gloves, and this is not quantifiable. An analysis of the changes in the water source and disinfection was not the goal of this paper.
The study primarily focuses on microbiota shifts, but how do these correlate with host immune responses?
Response: We did not analyze the immune response when the microbiota transfers stopped working because we did not know that they had stopped working for many months after initiation of the experiment. Experiments evaluating the immune response, now that the male transfers are once again consistently effective, are ongoing, but are beyond the scope of this study since they do not involve conditions in which the microbiota-mediated protection was not effective since the specific conditions that lead to that were not identified. One of our studies that evaluated a component of the immune response has already been published (reference 58).
Have cytokine levels, immune cell populations, or other immune markers been analyzed to confirm that disease suppression is microbiota-driven rather than influenced by other
physiological changes?
Response: Transfer of male, but not female, microbiota into intact female recipients delays disease and mortality. By definition, the disease suppression is microbiota-driven. Experiments evaluating the immune response, now that the male transfers are once again consistently effective, are ongoing, but are beyond the scope of this study. One of our studies that evaluated a component of the immune response has already been published (reference 58).
Could a germ-free or antibiotic-treated mouse model, recolonized with protective microbiota, confirm causality?
Response: In this manuscript, we have reported on our observation in normal mice that have intact microbiota and we find that male, but not female, microbiota transplants protect female mice from disease under those conditions. We are not sure how performing the same experiments in germ-free or antibiotic-treated mice will provide any further information. Those experiments are beyond the scope of this study.
Investigating whether fecal microbiota transplantation (FMT) or specific bacterial consortia can replicate the protective effect could enhance translational potential.
Response: Experiments using specific bacterial species are ongoing and are beyond the scope of this study.
Comments on the Quality of English Language: A revision of the entire text is required.
Response: The manuscript has been significantly revised in response to this and other reviewers' comments.

Reviewer 5 Report
Comments and Suggestions for Authors
- The Materials and Methods section is very confusing and needs more detail and clarification. There are no details about the number and weight of mice and the duration of the study. You will find some details in the Results section, and it is better to arrange and explain all the details of the methods better in the Materials and Methods section.
- There are not enough details about the method of collecting and preparing feces, whether for DNA analysis or transportation.
- How was urine collected and protein estimated?
- Do the results, lines 150 and 151, pertain to the study or to references 13 and 14? Please explain.
- The discussion part needs more in-depth explanations and no repetition to explain the results. There are also some studies that have addressed the idea of transferring microbiota, although not from resistant males to more susceptible females. These studies can be referred to and their results compared to the results of the present study.
- What is the final conclusion of the study and what recommendation can be made through the study about development of therapies for the treatment of SLE?

Author Response
Review #5
Response: We would like to thank the reviewer for taking the time to provide a thoughtful review of our manuscript. We believe the resulting revisions will significantly strengthen the manuscript.
The Materials and Methods section is very confusing and needs more detail and clarification.
Response: We apologize for the confusion. The Methods and Materials section has been extensively revised in response to this and other reviewers’ comments.
There are no details about the number and weight of mice and the duration of the study.
Response: The animal numbers (n) were included in the legend for each figure, but that information has now also been added to the Materials and Methods section in section 2.5 for the kidney disease experiments (L186-189) and in section 2.6 (L191-196) for the microbiota analysis. The weight average and range of the donor mice and weanlings has been added to sections 2.3 (L140-141) and 2.4 (L154), respectively. We did not weigh the experimental mice during the disease experiments because in our experience with BWF1 mice in our colony, weight loss does not correlate with disease progression. We are not sure what the reviewer means by "the duration of the study" but have attempted to provide some detail about duration. The duration of the disease experiments varies and is indicated in the figures showing the disease data. The cecal transplant protocol is described in more detail in section 2.4 and includes a more detailed description of the different feeding time-points (L159-164). The microbiota analysis is now described in more detail in sections 2.6 and was performed on feces from 16-week-old female and male mice, and on feces from female or male mice that had received cecal contents 4 weeks earlier at 15/16 weeks of age and/or 19 weeks of age (L191-196), and are also indicated in the legends for each figure.
You will find some details in the Results section, and it is better to arrange and explain all the details of the methods better in the Materials and Methods section.
Response: We apologize for the confusion. The Material and Methods section has been extensively revised in response to this and other reviewers' comments.
There are not enough details about the method of collecting and preparing feces, whether for DNA analysis or transportation.
Response: The Material and Methods section has been extensively revised in response to this and other reviewers. That information is now located in sections 2.2, 2.3, and 2.4.
How was urine collected and protein estimated?
Response: We apologize for the omission. The criteria for kidney disease is now described in new section 2.5.
Do the results, lines 150 and 151, pertain to the study or to references 13 and 14? Please
explain.
Response (L239-242): We apologize for the confusion. We have modified the statement to indicate the age at which female mice begin developing disease and die in our facility. The statement about male mice developing a milder form of disease refers to male mice in our colony and in several of the early studies on the BWF1 model of SLE (references have been modified).
The discussion part needs more in-depth explanations and no repetition to explain the results.
Response: The manuscript has been extensively revised and we have eliminated the repetition from the results section in response to this and other reviewers' comments. We have attempted to provide some in-depth explanations, but would need more specific guidance from the reviewer on where in-depth explanations are required.
There are also some studies that have addressed the idea of transferring microbiota, although not from resistant males to more susceptible females. These studies can be referred to and their results compared to the results of the present study.
Response (L722-732): Discussion of relevant study has been included in the revised manuscript.
What is the final conclusion of the study and what recommendation can be made through the study about development of therapies for the treatment of SLE?
Response (L740-756): Please see revised discussion.

Round 2
Reviewer 3 Report
Comments and Suggestions for Authors
Authors have improved quality of the presentation, that in fact was principle point asking for revision.
Reviewer 4 Report
Comments and Suggestions for Authors
The authors have addressed the suggestions and clarified key points that were highlighted. These adjustments have significantly enhanced the clarity of the study. I am satisfied with their responses.
Reviewer 5 Report
Comments and Suggestions for Authors
The manuscript has been reviewed, taking into account the proposed amendments, and the questions raised have been answered.